# Application of the ESMO Magnitude of Clinical Benefit Scale to assess the clinical benefit of antibody drug conjugates in solid cancer: a systematic descriptive analysis of phase III and pivotal phase II trials

Lieming Ding, Xiaobin Yuan, Yang Wang, Zhilin Shen  , Pengxiang Wu

LD and XY contributed equally.

Betta Pharmaceuticals Co Ltd, Hangzhou, China

**Correspondence to**
Dr Lieming Ding;
lieming.ding@bettapharma.com

## ABSTRACT

**Objective** The aim of this study was to assess the clinical benefit value of approved antibody drug conjugates (ADCs) for solid tumours using the European Society for Medical Oncology Magnitude of Clinical Benefit Scale (ESMO-MCBS) V.1.1.

**Design** Systematic descriptive analysis.

**Data sources** PubMed was searched for publications from 1 January 2000 to 18 October 2023.

**Eligibility criteria** We included the phase III randomised controlled trials or phase II pivotal trials leading to approval of ADCs in solid tumours.

**Data extraction and synthesis** Two independent reviewers extracted data and discrepancies were resolved by consensus in the presence of a third investigator.

**Results** ESMO-MCBS Scores were calculated for 16 positive clinical trials of eight ADCs, which were first approved by the US Food and Drug Administration (FDA), the European Medicines Agency (EMA), the China National Medical Products Administration and the Japanese Pharmaceuticals and Medical Devices Agency for solid cancers. Among 16 trials, 4 (25%) met the ESMO-MCBS benefit threshold grade, while 12 (75%) of the regimens did not meet the ESMO-MCBS benefit threshold grade. 5 (31%) of the 16 trials had no published scorecard on the ESMO website due to the approval by other jurisdictions but not by the FDA or EMA. Discrepancies between our results and the ESMO scorecard were observed in 4 (36%) of 11 trials, mostly owing to integration of more recent data.

**Conclusions** ESMO-MCBS is an important tool for assessing the clinical benefit of cancer drugs, but not all drugs met the meaningful benefit threshold.

## INTRODUCTION

Cancer is one of the leading causes of the death worldwide and ranks second in mortality below heart disease in the USA.[1 2] Considering the off-target toxicities of chemotherapeutic drugs, novel anticancer agents which can selectively target

## STRENGTHS AND LIMITATIONS OF THIS STUDY

⇒ This analysis assessed the clinical benefit in phase III randomised controlled trials or phase II pivotal trials leading to approval.

⇒ The European Society for Medical Oncology Magnitude of Clinical Benefit Scale (ESMO-MCBS) is a validated tool for grading the clinical benefit of novel drugs.

⇒ Shortcomings of the ESMO-MCBS were identified, which should be addressed in future versions.

⇒ The data lack some important information such as the drug costs analysis and American Society of Clinical Oncology Value Framework.

cancer cells and attenuate side effects are urgent. Antibody drug conjugates (ADCs) are the emerging next generation therapeutics after monoclonal antibodies, attributing to the superior anticancer activity over traditional chemotherapy.[3 4] In 2000, the US Food and Drug Administration (FDA) approved the first ADC, gemtuzumab ozogamicin (Mylotarg), for the treatment of patients with acute myeloid leukaemia.[5] To date, a total of 15 ADCs have been approved worldwide and more than 100 ADCs are currently being evaluated in clinical trials.[6]

The paradigm of cancer therapy has shifted from the disease-centred strategy to a patient-centred strategy, emphasising the comprehensive value of the therapeutic regimen, including quality of life (QoL) and cost. However, it is always hard to objectively evaluate therapy value and clinical benefit. Both the American Society of Clinical Oncology Value Framework[7 8] and the European Society for Medical Oncology Magnitude of Clinical Benefit Scale (ESMO-MCBS)[9 10]

aimed to evaluate the clinical benefit of novel anticancer treatments. Besides, the ESMO-MCBS seems to be very reliable in advanced or metastatic diseases across all therapeutic settings in daily practice.[11]

Randomised controlled trials are regarded as the 'gold standard' in evaluating medicinal products.[12] Considering an unmet medical need, single-arm trials can also support approvals, although uncommon. Here, we overview the landscape of ADCs approved by the FDA, the European Medicines Agency (EMA), the China National Medical Products Administration (NMPA) and the Japanese Pharmaceuticals and Medical Devices Agency (PMDA) for solid cancer, and describe the clinical benefit of these ADCs and further explore whether clinical benefit is related to high-quality cancer care.

## METHODS
### Design
Our study is a systematic descriptive analysis. There was no protocol and the study was not registered.

### Data sources and searches
We identified all approved indications of eight ADCs between 1 January 2000 and 18 October 2023 by searching different official websites, including FDA (https://www.fda.gov/), EMA (http://www.ema.europa.eu), NMPA (https://www.nmpa.gov.cn/) and PMDA (https://www.pmda.go.jp/english/index.html). Drug names, indications and approval dates were recorded from these official websites. In addition, PubMed (https://pubmed.ncbi.nlm.nih.gov/) was searched for publications from 1 January 2000 to 18 October 2023. The full search terms used in this study are presented in online supplemental table S1. The search was limited to human studies and had no language restrictions.

### Study selection
The inclusion criterions were the phase II and phase III randomised controlled trials that led to approval by regulatory authorities. The phase II single-arm trials were included if they were pivotal to obtain its marketing approval. Updated data of pivotal studies or reports on toxicity or QoL outcomes were also considered.

The exclusion criteria were: non-pivotal trials (eg, the secondary, subset, meta-analyses or systematic reviews; phase I or IV trials; exploratory trials); prematurely stopped randomised controlled trials; no drug therapy intervention; haematological cancers; non-English articles.

### Data extraction
The study was reviewed independently by two investigators (XY and YW) and relevant details were extracted from each study, including study design, sample size and endpoints (objective response rate (ORR), disease-free survival (DFS), progression free survival (PFS), overall survival (OS), QoL and toxicity). The discrepancies were resolved by consensus in the presence of a third investigator (LD).

### ESMO-MCBS scoring
The ESMO-MCBS can be used to rank the value of systemic therapies based on reported relative and absolute benefits in terms of improved survival (DFS, PFS and OS) and better survival (eg, QoL, toxicity). It is worth noting that the ESMO-MCBS can only be applied to studies showing statistically significant differences. For the non-curative setting, form 2a and form 2b are available, considering the predefined primary and secondary endpoints with regard to absolute gain as well as the lower end of the 95% CI of the corresponding HR. In the context of non-inferiority trials, form 2c has been devised, incorporating QoL and toxicity data to evaluate scores. Additionally, form 3 is designated to assess single-arm studies. Initial score was adjusted based on various ESMO criteria, including long-term survival, toxicity and QoL.[13] Palliative therapies were ultimately graded 1–5 for advanced disease setting and as A, B or C for adjuvant or neoadjuvant therapy setting, with scores 5, 4, A or B representing substantial clinical benefit.[14]

### Patient and public involvement
None.

## RESULTS
### Trial selection and characteristics
After excluding 7 of 15 approved ADCs (ie, gemtuzumab ozogamicin, brentuximab vedotin, inotuzumab ozogamicin, moxetumomab pasudotox, polatuzumab vedotin, belantamab mafodotin and loncastuximab tesirine) for haematological cancers, clinical trials on 8 ADCs for solid cancers were identified. A total of 21 publications from 16 clinical trials (nine randomised controlled trials and seven single-arm trials) leading to approval for solid cancers were finally identified, including 2 for trastuzumab emtansine, 1 for enfortumab vedotin, 5 for trastuzumab deruxtecan, 3 for sacituzumab govitecan, 1 for cetuximab saratolacan, 2 for disitamab vedotin, 1 for tisotumab vedotin and 1 for mirvetuximab soravtansine. The approved ADCs covered seven solid cancer types, including breast cancer, urothelial carcinoma, gastric cancer, non-small cell lung cancer, head-and-neck squamous cell carcinoma, cervical cancer and ovarian cancer (table 1). The publications on updated efficacy or QoL data were also included for ESMO-MCBS score adjustment.[15–35]

### ESMO-MCBS Scores
In total, 16 clinical trials were assessed by form 1 (n=1), form 2a (n=6), form 2c (n=1) and form 3 (n=8). Among 16 trials, 4 (25%) of the regimens met the ESMO-MCBS benefit threshold grade, while 12 (75%) of the regimens did not meet the ESMO-MCBS benefit threshold grade (table 2).

**Table 1** Characteristics of the included studies

| Drug name | Trial name/Registry number | Design | Indications | Approval data | Ref |
|---|---|---|---|---|---|
| Trastuzumab emtansine | KATHERINE NCT01772472 | Phase III RCT | HER2-positive early breast cancer after receiving neoadjuvant therapy containing a taxane (with or without anthracycline) and trastuzumab | 2019 by FDA 2019 by EMA | 15 |
| | EMILIA NCT00829166 | Phase III RCT | HER2-positive advanced breast cancer second line | 2013 by FDA 2013 by EMA | 16–18 |
| Enfortumab vedotin | EV-301 NCT03474107 | Phase III RCT | Locally advanced or metastatic urothelial carcinoma | 2019 by FDA 2022 by EMA | 19 |
| Trastuzumab deruxtecan | DESTINY-Breast01 NCT03248492 | Phase II SAT | HER2-positive metastatic breast cancer third line | 2019 by FDA 2020 by EMA | 20 |
| | DESTINY-Breast03 NCT03529110 | Phase III RCT | HER2-positive metastatic breast cancer second line | 2022 by FDA 2022 by EMA 2023 by NMPA | 21 22 |
| | DESTINY-Breast04 NCT03734029 | Phase III RCT | HER2-low metastatic breast cancer | 2022 by FDA 2022 by EMA 2023 by NMPA | 23 |
| | DESTINY-Gastric01 NCT03329690 | Phase II RCT | HER2-positive advanced gastric or gastro-oesophageal junction adenocarcinoma after a prior trastuzumab-based regimen | 2021 by FDA 2022 by EMA | 24 |
| | DESTINY-Lung02 NCT04644237 | Phase II SAT | Metastatic HER2-mutant NSCLC second line | 2022 by FDA | 25 |
| Sacituzumab govitecan | ASCENT NCT02574455 | Phase III RCT | Relapsed or refractory metastatic triple-negative breast cancer | 2020 by FDA 2021 by EMA 2022 by NMPA | 26 27 |
| | TROPiCS-02 NCT03901339 | Phase III RCT | Endocrine-resistant, chemotherapy-treated HR+/HER2– locally recurrent inoperable or metastatic breast cancer | 2023 by FDA 2023 by EMA | 28 29 |
| | TROPHY-U-01 NCT03547973 | Phase II RCT | Metastatic urothelial carcinoma | 2021 by FDA | 30 |
| Cetuximab saratolacan | – NCT02422979 | Phase II SAT | Recurrent head and neck squamous cell carcinoma | 2020 by PMDA | 31 |
| Disitamab vedotin | – NCT03556345 | Phase II SAT | HER2-overexpressing, locally advanced or metastatic gastric or gastro-oesophageal junction cancer | 2021 by NMPA | 32 |
| | – NCT03507166 | Phase II SAT | HER2+ locally advanced or metastatic urothelial carcinoma | 2022 by NMPA | 33 |
| Tisotumab vedotin | innovaTV 204 NCT03438396 | Phase II SAT | Recurrent or metastatic cervical cancer | 2021 by FDA | 34 |
| Mirvetuximab soravtansine | SORAYA NCT04296890 | Phase II SAT | FRa-high, platinum-resistant epithelial ovarian cancer | 2022 by FDA | 35 |

EMA, European Medicines Agency; FDA, Food and Drug Administration; HRQoL, health-related quality of life; NMPA, National Medical Products Administration; NSCLC, non-small cell lung cancer; PMDA, Japanese Pharmaceuticals and Medical Devices Agency; RCT, randomised control trial; SAT, single-arm trial.

In trials meeting the ESMO threshold for clinically meaningful benefit (scores 4–5), the median PFS and median OS seemed slightly extended in contrast to those falling below the threshold (scores 1–3), despite that statistically significant differences were not calculated.

On the comparison of our final ESMO-MCBS Scores with the relevant scorecards accessible through the ESMO website, 5 out of 16 (31%) trials did not yet have the published scorecard, mostly because they are approved by jurisdictions other than the EMA and FDA. In our study,

**Table 2** Clinical benefit according to ESMO-MCBS V.1.1 and concordance rate between ESMO Framework in this study and ESMO-MCBS scorecards

| Trial name | Intervention group | PO | PO control group | PO gain | PO HR (95% CI) | Adjustments | ESMO-MCBS V.1.1 | Scorecards |
|---|---|---|---|---|---|---|---|---|
| KATHERINE | T-DM1 | DFS | 3-year: 77.0% | 3-year: 11.3% | 0.50 (0.39 to 0.64) | NA | A (form 1) | A |
| EMILIA | T-DM1 | PFS and OS | 25.9 months | 4.0 months | 0.75 (0.64 to 0.88) | Improved HRQoL | 3 (form 2a) | 4 |
| EV-301 | EV | OS | 8.97 months | 3.91 months | 0.70 (0.56 to 0.89) | NA | 4 (form 2a) | 4 |
| DESTINY-Breast01 | DS-8201 | ORR | – | – | – | NA | 2 (form 3) | 2 |
| DESTINY-Breast03 | DS-8201 | PFS (OS improved) | 2-year: 69.9% | 2-year: 7.5% | 0.64 (0.47 to 0.87) | NA | 3 (form 2a) | 4 |
| DESTINY-Breast04 | DS-8201 | PFS (OS improved) | 16.8 months | 6.6 months | 0.64 (0.49 to 0.84) | NA | 4 (form 2a) | 4 |
| DESTINY-Gastric01 | DS-8201 | ORR | 14% | 37% | – | NA | 2 (form 2c) | 4 |
| DESTINY-Lung02 | DS-8201 | ORR | – | – | – | NA | 3 (form 3) | NA |
| ASCENT | SG | PFS (OS improved) | 6.9 months | 4.9 months | 0.51 (0.41 to 0.62) | Increased toxicity/ improved HRQoL | 4 (form 2a) | 4 |
| TROPiCS-02 | SG | PFS (OS improved) | 11.2 months | 3.2 months | 0.79 (0.65 to 0.96) | NA | 1 (form 2a) | 3 |
| TROPHY-U-01 | SG | ORR | – | – | – | NA | 3 (form 3) | NA |
| NCT02422979 | CS | ORR | – | – | – | NA | 2 (form 3) | NA |
| NCT03556345 | DV | ORR | – | – | – | NA | 2 (form 3) | NA |
| NCT03507166 | DV | ORR | – | – | – | NA | 3 (form 3) | NA |
| innovaTV 204 | TV | ORR | – | – | – | NA | 2 (form 3) | 2 |
| SORAYA | MS | ORR | – | – | – | NA | 2 (form 3) | 2 |

CS, cetuximab saratolacan; DFS, disease-free survival; DV, distitamab vedotin; EV, enfortumab vedotin; MS, mirvetuximab soravtansine ORR, objective response rate; OS, overall survival; PFS, progression-free survival; PO, primary outcome; SG, sacituzumab govitecan; T-DM1, trastuzumab emtansine; TV, tisotumab vedotin.

several ADCs have been only approved by local agencies in China and Japan. Discrepancies were observed in 4 of 11 (36%) trials (table 2). We find different scores mostly due to integration of more recent data. When it comes to multiple publications, the clinical benefit was evaluated based on both the primary and secondary manuscripts. In our study, we presented the scores based on the newly published articles due to the more mature OS data or QoL data. Interestingly, discrepancies were observed in EMILIA and TROPiCS Trials. The scores in EMILIA based on the interim analyses and final OS analyses were 4 and 3, respectively. The score was decreased as the OS HR was increased. Similarly, the scores in TROPiCS-02 based on the interim analyses and final OS analyses were 3 and 1, respectively. As OS data were mature and showed benefit, the form 2b was replaced by form 2a; however, the OS HR was increased, leading the low score. Besides, in the DESTINY-Breast03 Study, given that the form 2b was replaced by form 2a, we ignored the long-term plateau in the PFS curve, resulting in the discrepancy. According to the ESMO-MCBS scorecard, DESTINY-Gastric01 was evaluated as 4 with the form 2a instead of form 2c, and the main reason was that the hierarchical primary outcome was OS.

## DISCUSSION

To the best of our knowledge, this is the first review to apply ESMO-MCBS V.1.1 exclusively to approved ADCs. In this study, we used the ESMO-MCBS tool to gauge the magnitude of clinical benefit derived from phases II and III trials on ADC therapy for solid tumours. Our cohort of 16 clinical trials encompassed multiple therapeutic options with ADCs from 2000 to 2023.

Despite the preferred trials for the application of marketing approval, randomised controlled trials are not always likely to perform. In recent years, various drugs have been approved based on the single-arm trials. These approvals were instituted to fulfil an unmet medical necessity, with less comprehensive data compared with those for standard approval processes. However, the evidence level that the investigational drugs provide clinical benefits may be insufficient when solely based on single-arm trials. In our study, to evaluate whether the authorised drugs based on single-arm trials provided substantial benefit, we included these pivotal phase II single-arm trials leading to the approval.

The application of ESMO-MCBS forms on the basis of OS and/or PFS was less than those based on ORR (38% vs 56%). In five trials with the primary endpoint of PFS, four (80%) trials were assessed using form 2a instead of form 2b due to the OS advantage. However, a previous study revealed that conclusions based on PFS outcomes in these clinical trials might be overestimated by this scoring system, due to the statistical power to only detect significant differences in PFS, rather than OS.[14] ORR is not a direct measure of clinical benefit; however, it is a measure of anticancer activity.[36] A study found that 38% and 34%

of anticancer drugs were approved by FDA based on ORR and PFS, shortening the development durations by 19 months and 11 months, respectively.[37] Nevertheless, among 93 anticancer drugs with accelerated approvals, only 20% demonstrated an improvement in OS in confirmatory trials.[38] As previously revealed, the definition of clinical value is different between stakeholders, causing different conclusions.[39] For instance, the ESMO considers benefit as 'living longer and/or living better', which resonates in the ESMO-MCBS form for single-arm trials.[9 10] This is confirmed in our findings that the benefit of the majority of ADCs was 'modest' according to ESMO-MCBS Scores. A high ESMO-MCBS Score for single-arm trials was related to favourable efficacy in combination with an improved QoL.[40] Yet, delayed publications or publication bias for QoL is overlooked by ESMO-MCBS.[41] Therefore, QoL appears of lesser importance in regulatory decision making on single-arm trials.

Despite a minor number of inconsistencies between our final scores and the relevant ESMO scorecards, the final scores varied significantly due to these disparities. The reassessment of the ESMO-MCBS Score is important when results from confirmatory trials are published, which may affect the ESMO-MCBS Score. Furthermore, extended follow-up may also affect the EMSO-MCBS Score. For example, we previously assigned an ESMO-MCBS Score of '3' for sacituzumab govitecan (ASCENT Trial) due to the increased toxicity; however, we finally assigned a score of '4' due to a recent publication on health-related QoL.[27] On the contrary, the score was found to decrease as OS data were mature and showed benefit while the value of OS HR was increased in the EMILIA[16 17] and TROPiCS Trials.[28 29] ESMO-MCBS can develop a specialised form for re-evaluation of the updated OS data. Additionally, adjustments sometimes might be confusing. For example, we assigned an ESMO-MCBS Score of '3' for trastuzumab deruxtecan (DESTINY-Breast03 Trial) based on form 2a due to the improved OS; however, the ESMO-MCBS scorecard showed '4' due to the long-term plateau in the PFS curve based on form 2b. Although grades ≥3 adverse events may be different between the treatment arms, it was not always statistically significant in publications. Besides, adverse events that can affect the daily well-being of patients may not be consistently published. Clearer definitions or quantifications of toxicity profiles in future ESMO-MCBS versions are highly desirable. Furthermore, several trials had no published scorecard, because they are approved by jurisdictions other than the EMA and FDA. We suggest that ESMO-MCBS scorecards can also publish the drugs that are approved by other agencies other than FDA and EMA.

Almost half of positive clinical trials are unable to demonstrate a substantial clinical benefit based on the ESMO framework, and this has drawn criticism in light of market approvals for anticancer drugs in recent periods.[42–44] In our study, only 4/16 (25%) of the clinical trials can meet the ESMO-MCBS threshold for clinical benefit. Notably, all single-arm trials cannot meet

the ESMO-MCBS threshold for clinical benefit. Form 3 is used to assess single-arm trials, but it should also be noted that by using this form, only grades from 1 to 4 can be attained. As a result, the threshold can only be met if the study demonstrates HRQoL improvements, or additionally, if confirmatory phase IV trials are accessible.

Furthermore, in a head-to-head trial (DESTINY-Breast03) to compare the efficacy and safety of trastuzumab deruxtecan with those of trastuzumab emtansine in patients with HER2-positive metastatic breast cancer, trastuzumab deruxtecan showed a superior OS over trastuzumab emtansine.[21 22] Thus, we are also confused that whether the OS gain in the ESMO-MCBS tool can evaluate the benefits of different drugs in the same indication or the same drug in different indications. The thresholds for the OS gain appear a bit lenient.

This study has several implications. First, this study showed clinical benefits for limited ADCs in solid tumours, suggesting that subsequent clinical trials on the treatment of solid tumours with ADCs should follow the cases with meaningful clinical benefit. Second, value frameworks can help not only identify drugs with low or uncertain clinical benefit that should be targeted for price negotiations, but also therapies with evidence of higher clinical benefit to improve access to benefit drugs, thereby contributing to patient-centred cancer treatment goals. Finally, the ADCs with HRQoL improvement showed clinical benefit, suggesting that HRQoL should be paid sufficient attention in clinical trials and clinical treatment strategies.

There are several limitations in our study. First, we limited our analysis to solid tumours and excluded drugs approved to treat haematological malignancies. When this study was completed in June 2023, the haematology-specific version has yet to be published. We will further access the clinical benefits of ADCs for haematological malignancies according to the currently published ESMO-MCBS:H V.1.0.[45] Another main limitation is publication selection bias. However, we implemented a rigorous search strategy to mitigate it. Publications were selected by two investigators independently based on predefined inclusion and exclusion criteria. We specifically excluded phase I and non-pivotal phase II trials to evaluate novel therapies with sufficient efficacy and toxicity data, thereby primarily including authorised regimens applied in routine clinical practice. Furthermore, the limited toxicity data accessible within published clinical trials hindered our ability to accurately adjust preliminary grades. Additionally, our study was also limited by the lack of drug costs, which may help clinicians to select the optimal therapeutic drugs developed for the same clinical entity.[23]

In conclusion, this is the first study that used the ESMO-MCBS tool to assess the clinical benefit of ADCs across several solid cancers. ESMO-MCBS are important tools for assessing the clinical benefit of cancer drugs, but not all drugs met the meaningful benefit threshold. Those therapeutic regimens with improved HRQoL showed clinical value, suggesting that clinical trials and clinical treatment strategies should pay more attention to HRQoL. Furthermore, a more exhaustive rule for toxicity penalties due to adverse event is required, as well as the approach to an adjusted scoring for the trials with HRQoL data.

**Contributors** LD, YW and XY designed the study. XY, ZS and PW extracted the data from all sources, performed the analyses and drafted the manuscript. All authors critically revised the manuscript. LD is responsible for the accuracy of the data and accepts full responsibility for the work and/or the conduct of the study, had access to the data, and controlled the decision to publish.

**Funding** The authors have not declared a specific grant for this research from any funding agency in the public, commercial or not-for-profit sectors.

**Competing interests** None declared.

**Patient and public involvement** Patients and/or the public were not involved in the design, or conduct, or reporting, or dissemination plans of this research.

**Patient consent for publication** Not applicable.

**Ethics approval** Not applicable.

**Provenance and peer review** Not commissioned; externally peer reviewed.

**Data availability statement** Data are available upon reasonable request.

**ORCID iD**
Zhilin Shen http://orcid.org/0000-0003-4128-4524

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
