## [Reviewer comments · BMJ Open]

ARTICLE DETAILS

TITLE (PROVISIONAL)	Application of the ESMO Magnitude of Clinical Benefit Scale to assess the clinical benefit of antibody–drug conjugates in solid cancer: a systematic descriptive analysis of phase 3 and pivotal phase 2 trials
AUTHORS	Ding, Lieming; Yuan, Xiaobin; Wang, Yang; Shen, Zhilin; Wu, Pengxiang

VERSION 1 – REVIEW

REVIEWER	Grossmann, Nicole Austrian Inst Hlth Technol Assessment
REVIEW RETURNED	22-Aug-2023

GENERAL COMMENTS	This is a well-written manuscript that needs to undergo a few minor changes. The suggested changes are listed below. Introduction: Lines 42-44 state that 13 drugs have been approved by the FDA. It would be interesting why five of these drugs are not included in this study. In addition, if the approval of ADCs of different agencies is relevant for this paper the information on all agencies and their current number of approved ADCs would be relevant. Lines 56-60 state that the landscape of ADCs approved by several agencies is outlined in the paper. However, it is not mentioned in the results or in Table 2 where the drugs have been approved (by all agencies?). If this was not part of the study the aim of the introduction should be adapted accordingly. Methods: The flow diagram should include information on why the studies were not applicable to be graded by the ESMO-MCBS. In addition, information on the study design (number of RCTs and single-arm trials) of the included studies would be valuable. The ESMO-MCBS can only be applied to studies showing statistically significant differences. This information should be mentioned in the methods section. Line 82 states that Form 3 is used to assess single-arm trials, but it should also be noted that by utilizing this form, only grades from 1 to 4 can be attained. As a result, the threshold can only be met if the study demonstrates quality of life (QoL) improvements, or additionally, if confirmatory phase 4 trials are accessible. Results: The results are only briefly described. The information regarding the study design should be included, such as the number of single-arm
--

	trials and randomized controlled trials (RCTs). Additionally, it could be interesting to include the number of different forms that were utilized. Details such as the study phase, year of publication, and study design should also be incorporated into Table 2. Merging Tables 2 and 3 could provide a clearer overview of the results. The table abbreviations should be placed above the tables. In the case of the ASCENT trial, if there were alterations to the preliminary ESMO-MCBS score, this information could be included directly in the table or linked with a description provided above. Which information was utilized when multiple publications were available for a single study? Or were there never any discrepancies? The discrepancies concerning the scorecards and the present studies could be described in greater detail in the results section. The authors mention that grades for only 11 studies were available on the ESMO-MCBS Scorecards. This could be an additional point in the discussion since the ESMO states on its website (https://www.esmo.org/guidelines/esmo-mcbs/esmo-mcbs-for-solid-tumours/esmo-mcbs-scorecards?mcbs_score_cards_form%5BsearchText%5D=T-DM1) that the Scorecards include all cancer medicines approved by the European Medicines Agency (from January 2016) and the US Food and Drug Administration (from January 2020). Why are the remaining 8 studies not part of the Scorecards? The sentence of line 111 (4 of them on experimental therapies that had granted the market authorization, and continue to be authorized for human use) needs clarification in my point of few since it's not entirely clear what is meant by that. Trastuzumab deruxtecan is currently not approved by any agency according to Table 2. Is it currently under evaluation since it may be a recently published study or is there another reason? This information should be added to the paper.
--	---

REVIEWER	Kiesewetter, Barbara Medical University of Vienna
REVIEW RETURNED	03-Sep-2023

GENERAL COMMENTS	Although the methodology is concise and clear, in my opinion, this work does not offer any relevant novelties or valuable insights. In particular, the applied procedure of re-evaluating and summarizing the scores reviewed by the ESMO-MCBS team seems to me not very useful. In some studies, the authors find different scores than the ESMO team, which is mostly due to integration of more recent data (not yet considered by ESMO). It is of course also true that MCBS scoring has some problems that have been published in the past (Gyawali B 2021), and are currently subject to revision of the score and it therefore makes sense to assess its applicability specifically for new compounds such as ADCs, but the approach taken here does not really profoundly answer this question.
--

REVIEWER	Tibau, Ariadna Sant Pau and Universitat Autònoma de Barcelona
REVIEW RETURNED	05-Sep-2023

GENERAL COMMENTS

The manuscript aims to assess the clinical benefit of approved antibody-drug conjugates (ADCs) for solid tumors using the European Society for Medical Oncology Magnitude of Clinical Benefit Scale (ESMO-MCBS) v1.1.

Having said that, my major issue with this paper is the lack of clarity by the authors on the key points they are trying to make.

A few specific comments follow:

1) Title Clarity: The manuscript's title could be more informative and specific to the research topic. In the context of cancer, "value" typically refers to the assessment of the clinical benefit that a specific cancer treatment or intervention provides relative to its cost. However, it's important to note that the authors did not evaluate the cost of cancer drugs in this study. Therefore, substituting the word "value" with "clinical benefit" in the title may provide a more accurate representation of the study's focus.

2) The abstract lacks critical information, such as the number of specific ADCs evaluated and the jurisdictions in which they are approved. It would be beneficial to include a concise summary of the implications of this study in the conclusion section of the abstract.

3) In this study, the concept of indications/approvals, trials and tumors is very confusing. On one hand, in the introduction, the authors state, "we overview the landscape of ADCs approved by FDA, European Medicines Agency (EMA), National Medical Products Administration (NMPA), and Japanese Pharmaceuticals and Medical Devices Agency (PMDA) for solid cancer between 2000 and 2023." However, in the methods and results sections, it is specified how this information was collected, including approval dates and the indications approved for each jurisdiction. Later on, the authors claim: "The approved ADCs covered 7 indications, including breast cancer, urothelial carcinoma, gastric cancer, non-small cell lung cancer, head-and-neck squamous cell carcinoma, cervical cancer, and ovarian cancer (table 1)." This is not possible because both T-DM1 and trastuzumab deruxtecan in breast cancer have multiple indications. Therefore, the total number of indications cannot be equivalent to the number of tumors involved. This part of the study should be thoroughly reviewed and clarified.

4) The authors should also clarify that the National Medical Products Administration (NMPA) pertains to China.

5) To the best of my knowledge, THERESA and DESTINY-Breast02 are not pivotal trials leading to the approval of cancer drugs, but I may be mistaken. Could the authors improve Table 1 by specifying in which jurisdictions these trials are considered pivotal and where these drugs are approved? Can the authors clarify how they have defined a study as pivotal?

6) Additionally, why was DESTINY-CRC01 included in Table 1 and as one of the 19 trials to be analyzed if trastuzumab deruxtecan is not approved for colorectal cancer, and non-approval was a criterion for exclusion in your study?

	7) While the authors state that they exclude non-RCTs, it appears that the DESTINY Breast01 trial (a single-arm trial) has been included in the study. Can the authors specify inclusion criteria and exclusion criteria more clearly? 8) The title of Table 3 could be improved to enhance clarity, such as 'Rate of Concordance Between ESMO Framework in this Study and ESMO-MCBS Scorecards.' Additionally, including a legend with abbreviations and incorporating the names of the evaluated drugs would enhance the table's comprehensibility. 9) The sentence on Page 6, Line 90, states, 'A flow chart indicating the selection procedure is shown in Figure 1. A total of 25 studies from 19 clinical trials were finally identified, including...'. However, it seems there is an inconsistency between the 25 studies mentioned in Figure 1 and the 19 clinical trials stated in the article. The authors should clarify this discrepancy. I suggest to make the distinction between studies (25) and approvals/indications (19) 10) Figure 1 shows that among the initially selected 34 trials, 9 studies (26%) were excluded because the ESMO-MCBS could not be applied. It would be beneficial if the authors could provide a more detailed explanation in the text regarding why the ESMO-MCBS could not be applied in these 9 studies. 11) ESMO-MCBS scorecards are only published for drugs that are approved by the FDA and EMA. Can the authors clarify if the 8 studies that do not have scorecards are because they are only approved by jurisdictions other than the European and American ones? In the case of one study (trastuzumab deruxtecan in colorectal cancer), it could be because it is not approved. 12) Limitations: While the manuscript mentions limitations, it would benefit from a more comprehensive discussion of potential biases and how they might affect the study's conclusions. For example, among the 13 therapeutic ADCs approved by the FDA, 7 belong to the hematological field. This is a major limitation that should be at least mentioned. 13) Interpretation of Results: The discussion could provide a deeper analysis of the findings and their clinical implications in relationship to ADCs. 14) Conclusion: The conclusion section is somewhat abrupt and could be expanded to provide a more comprehensive summary of the study's significance, public health implications and potential future directions
--	--

VERSION 1 – AUTHOR RESPONSE

Reviewer: 1

Dr. Nicole Grossmann, Austrian Inst Hlth Technol Assessment
Comments to the Author:

This is a well-written manuscript that needs to undergo a few minor changes. The suggested changes are listed below.

Introduction:

Lines 42-44 state that 13 drugs have been approved by the FDA. It would be interesting why five of these drugs are not included in this study. In addition, if the approval of ADCs of different agencies is relevant for this paper the information on all agencies and their current number of approved ADCs would be relevant.

Response: We apologize for this confusion. Five of 13 drugs are not included in this study because these five drugs (gemtuzumab ozogamicin, brentuximab vedotin, inotuzumab ozogamicin, moxetumomab pasudotox and polatuzumab vedotin) were approved for haematological malignancies. Up to now, a total of 15 ADCs were approved for both hematological malignancies (7 ADCs) and solid tumors (8 ADCs) worldwide. And we have revised these misleading statements (Page 7, Lines 132-136). We also pasted it here for your review:

After excluding 7 of 15 approved ADCs (i.e. gemtuzumab ozogamicin, brentuximab vedotin, inotuzumab ozogamicin, moxetumomab pasudotox, polatuzumab vedotin, belantamab mafodotin and loncastuximab tesirine) for haematological cancers, clinical trials on 8 ADCs for solid cancers were identified.

Lines 56-60 state that the landscape of ADCs approved by several agencies is outlined in the paper. However, it is not mentioned in the results or in Table 2 where the drugs have been approved (by all agencies?). If this was not part of the study the aim of the introduction should be adapted accordingly.

Response: Thank you for your constructive comments. We have added the corresponding first approval agencies in the Table 1.

Methods:

The flow diagram should include information on why the studies were not applicable to be graded by the ESMO-MCBS. In addition, information on the study design (number of RCTs and single-arm trials) of the included studies would be valuable.

Response: We appreciate your important comments. The flow diagram seems to be inconsistent with the formulation in the Result section. Therefore, we have removed the Figure 1. Also, we have added the number of 9 RCTs and 8 single-arm trials in the Results section (Page 7, Line 137).

The ESMO-MCBS can only be applied to studies showing statistically significant differences. This information should be mentioned in the methods section.

Response: Thank you for your helpful comment and we have added the sentence in the Methods section (Page 6, Lines 114-115).

Line 82 states that Form 3 is used to assess single-arm trials, but it should also be noted that by utilizing this form, only grades from 1 to 4 can be attained. As a result, the threshold can only be met if the study demonstrates quality of life (QoL) improvements, or additionally, if confirmatory phase 4 trials are accessible.

Response: Special thanks to you for your helpful comments. It is really true as you suggested that the threshold can only be met if the study demonstrates QoL improvements. We have added these sentences in the Discussion section (Pages 10-11, Lines 239-243).

Results:

The results are only briefly described. The information regarding the study design should be included,

such as the number of single-arm trials and randomized controlled trials (RCTs). Additionally, it could be interesting to include the number of different forms that were utilized. Details such as the study phase, year of publication, and study design should also be incorporated into Table 2. Merging Tables 2 and 3 could provide a clearer overview of the results. The table abbreviations should be placed above the tables. In the case of the ASCENT trial, if there were alterations to the preliminary ESMO-MCBS score, this information could be included directly in the table or linked with a description provided above.

Response: Thank you for your constructive comments. We have added the numbers of single-arm trials and RCTs (Page 7, Line 137). The different forms that were utilized have been mentioned in Table 2 and we have added the number of different forms in the Results section (Page 7, Lines 145-147). Besides, we also merged the Table 2 and Table 3 as you kindly suggested.

Which information was utilized when multiple publications were available for a single study? Or were there never any discrepancies?

Response: We regret any confusion that may have resulted. When comes to multiple publications, we always assessed the study based on the newly published articles due to the more mature OS data or QoL data. The discrepancies were observed in EMILIA and TROPiCS trials. The scores in EMILIA based on the interim analyses and final OS analyses were 4 and 3, respectively. The score was decreased as the OS HR was increased. Similarly, the scores in TROPiCS-02 based on the interim analyses and final OS analyses were 3 and 1, respectively. As OS data was mature and showed benefit, the form 2b was replaced by form 2a; however, the OS HR was increased, leading the low score. We have added these discrepancies in the Results section and Discussion section (Page 8, Lines 168-177).

The discrepancies concerning the scorecards and the present studies could be described in greater detail in the results section.

Response: We are grateful for the suggestion. As we respond above, we find different scores mostly due to integration of more recent data. And we have added it in the Results section and Discussion section (Page 8, Lines 167-168).

The authors mention that grades for only 11 studies were available on the ESMO-MCBS Scorecards. This could be an additional point in the discussion since the ESMO states on its website (https://www.esmo.org/guidelines/esmo-mcbs/esmo-mcbs-for-solid-tumours/esmo-mcbs-scorecards?mcbs_score_cards_form%5BsearchText%5D=T-DM1) that the Scorecards include all cancer medicines approved by the European Medicines Agency (from January 2016) and the US Food and Drug Administration (from January 2020). Why are the remaining 8 studies not part of the Scorecards?

Response: We apologize for this confusion and error. A total of 7 studies do not have scorecards. The drugs were not approved based on 4 trials (TH3RESA, DESTINY-Breast02, DESTINY-Lung01 and DESTINY-CRC01), which was inconsistent with the methods in our manuscript, and thus we have removed these 4 trials. We have revised the Methods section. Another 3 trials (NCT02422979, NCT03556345 and NCT03507166) do not have scorecards because they are approved by jurisdictions other than the EMA and US FDA.

The sentence of line 111 (4 of them on experimental therapies that had granted the market authorization, and continue to be authorized for human use) needs clarification in my point of view since it's not entirely clear what is meant by that.

Response: We are very sorry for our wrong statement and we have removed it.

Trastuzumab deruxtecan is currently not approved by any agency according to Table 2. Is it currently under evaluation since it may be a recently published study or is there another reason? This information should be added to the paper.

Response: We are very sorry for our mistake. The DESTINY-CRC01 trial should be excluded. We have double checked our whole content and revised the inclusion criteria in the Methods section.

Reviewer: 3

Dr. Barbara Kiesewetter, Medical University of Vienna

Comments to the Author:

Although the methodology is concise and clear, in my opinion, this work does not offer any relevant novelties or valuable insights. In particular, the applied procedure of re-evaluating and summarizing the scores reviewed by the ESMO-MCBS team seems to me not very useful. In some studies, the authors find different scores than the ESMO team, which is mostly due to integration of more recent data (not yet considered by ESMO). It is of course also true that MCBS scoring has some problems that have been published in the past (Gyawali B 2021), and are currently subject to revision of the score and it therefore makes sense to assess its applicability specifically for new compounds such as ADCs, but the approach taken here does not really profoundly answer this question.

Response: Thank you for your constructive comments. We apologize for the confusion caused by the ways we presented our data and would like to have an opportunity to clarify our insights.

Progress in the development of new cancer therapies must be sustainable and supported by clear evidence of true patient benefits. The experience in using a validated and unbiased approach to the grading of clinical benefit such as the ESMO-MCBS is well established for the evaluation of systemic therapies in solid tumor oncology. However, to the best of our knowledge, the publications on the ESMO-MCBS from China remain limited. We aimed to learn how to apply the ESMO-MCBS tool to assess the clinical benefit, even participate in the development of a version of the ESMO-MCBS for specific term, if possible. Although this is our first time using this scoring tool, we found several shortcomings or confusion of the ESMO-MCBS tool and we have added some superficial suggestions for your guidance.

1. As we reported in this manuscript, extended follow-up may affect the EMSO-MCBS score. However, further follow-up data on OS, PFS or HRQoL in most studies were not yet published at the time of approval. ESMO-MCBS can develop specialized form for re-evaluation of the updated OS data.

2. Although grades 3 or higher adverse events differ between treatment arms, the statistical significance is always not available in the publications. Besides, any adverse events of grade 3 or higher but not those affecting patients' daily well-being, as denoted in the ESMO-MCBS v1.1 forms, were published. Further versions of the ESMO-MCBS that can clearly defined or quantified toxicity profiles are highly expected and desirable.

3. Form 3 is used to assess single-arm trials, but it should also be noted that by utilizing this form, only grades from 1 to 4 can be attained. As a result, the threshold can only be met if the study demonstrates HRQoL improvements, or additionally, if confirmatory phase 4 trials are accessible.

4. In a head-to-head trial (DESTINY-Breast03) to compare the efficacy and safety of trastuzumab deruxtecan with those of trastuzumab emtansine in patients with HER2-positive metastatic breast cancer, trastuzumab deruxtecan showed a superior OS over trastuzumab emtansine. Therefore, we also confused that whether the OS gain in ESMO-MCBS tool can evaluate the benefits of different drugs in the same indication and whether trastuzumab deruxtecan showed higher clinical benefit than trastuzumab deruxtecan.

5. ESMO-MCBS scorecards are only published for drugs that are approved by the FDA and EMA. We think this may not contribute to adequate education and promotion worldwide. In our study, several ADCs have been approved by local agencies in China and Japan. We suggest that ESMO-MCBS scorecards can also published the drugs that are approved by other agencies other than FDA and EMA.

Reviewer: 4

Dr. Ariadna Tibau, Sant Pau and Universitat Autònoma de Barcelona

Comments to the Author:

The manuscript aims to assess the clinical benefit of approved antibody-drug conjugates (ADCs) for solid tumors using the European Society for Medical Oncology Magnitude of Clinical Benefit Scale (ESMO-MCBS) v1.1.

Having said that, my major issue with this paper is the lack of clarity by the authors on the key points they are trying to make.

Response: We apologize for the confusion. We have double checked our manuscript and revised the whole text.

A few specific comments follow:

1) Title Clarity: The manuscript's title could be more informative and specific to the research topic. In the context of cancer, "value" typically refers to the assessment of the clinical benefit that a specific cancer treatment or intervention provides relative to its cost. However, it's important to note that the authors did not evaluate the cost of cancer drugs in this study. Therefore, substituting the word "value" with "clinical benefit" in the title may provide a more accurate representation of the study's focus.

Response: We thank the reviewer for the very valuable comments. We have revised the title to 'Application of the ESMO Magnitude of Clinical Benefit Scale to assess the clinical benefit of antibody–drug conjugates in solid cancer: a systematic descriptive analysis of phase 3 and pivotal phase 2 trials'.

2) The abstract lacks critical information, such as the number of specific ADCs evaluated and the jurisdictions in which they are approved. It would be beneficial to include a concise summary of the implications of this study in the conclusion section of the abstract.

Response: We are grateful for the suggestion. We have added the number of specific ADCs evaluated and the approval jurisdictions in the Abstract. We also added the implication in the Conclusion section of Abstract.

3) In this study, the concept of indications/approvals, trials and tumors is very confusing. On one hand, in the introduction, the authors state, "we overview the landscape of ADCs approved by FDA, European Medicines Agency (EMA), National Medical Products Administration (NMPA), and Japanese Pharmaceuticals and Medical Devices Agency (PMDA) for solid cancer between 2000 and 2023." However, in the methods and results sections, it is specified how this information was collected, including approval dates and the indications approved for each jurisdiction. Later on, the authors claim: "The approved ADCs covered 7 indications, including breast cancer, urothelial carcinoma, gastric cancer, non-small cell lung cancer, head-and-neck squamous cell carcinoma, cervical cancer, and ovarian cancer (table 1)." This is not possible because both T-DM1 and trastuzumab deruxtecan in breast cancer have multiple indications. Therefore, the total number of

indications cannot be equivalent to the number of tumors involved. This part of the study should be thoroughly reviewed and clarified.

Response: We apologize for this error. We have double checked the whole manuscript and removed the indications. Also, we replaced the “studies” by “publications”.

4) The authors should also clarify that the National Medical Products Administration (NMPA) pertains to China.

Response: Thank you for your suggestion and we have added it.

5) To the best of my knowledge, THERESA and DESTINY-Breast02 are not pivotal trials leading to the approval of cancer drugs, but I may be mistaken. Could the authors improve Table 1 by specifying in which jurisdictions these trials are considered pivotal and where these drugs are approved? Can the authors clarify how they have defined a study as pivotal?

Response: We regret any confusion that may have resulted. It is really true as Reviewer suggested that THERESA and DESTINY-Breast02 are not pivotal trials leading to the approval of cancer drugs. We have removed them and added the corresponding approval agencies in the Table 1.

6) Additionally, why was DESTINY-CRC01 included in Table 1 and as one of the 19 trials to be analyzed if trastuzumab deruxtecan is not approved for colorectal cancer, and non-approval was a criterion for exclusion in your study?

Response: We apologize for this error and we have removed it.

7) While the authors state that they exclude non-RCTs, it appears that the DESTINY Breast01 trial (a single-arm trial) has been included in the study. Can the authors specify inclusion criteria and exclusion criteria more clearly?

Response: We apologize for this confusion. We have removed the non-RCTs.

8) The title of Table 3 could be improved to enhance clarity, such as 'Rate of Concordance Between ESMO Framework in this Study and ESMO-MCBS Scorecards.' Additionally, including a legend with abbreviations and incorporating the names of the evaluated drugs would enhance the table's comprehensibility.

Response: Thank you very much for your important comments and we have merged Table 2 and Table 3. And we added the abbreviations.

9) The sentence on Page 6, Line 90, states, 'A flow chart indicating the selection procedure is shown in Figure 1. A total of 25 studies from 19 clinical trials were finally identified, including...' However, it seems there is an inconsistency between the 25 studies mentioned in Figure 1 and the 19 clinical trials stated in the article. The authors should clarify this discrepancy. I suggest to make the distinction between studies (25) and approvals/indications (19)

Response: We apologize for this apparent confusion. We have removed the flow chart of the selection procedure and directly included the trials leading to approval. Also, we replaced the “studies” by “publications”.

10) Figure 1 shows that among the initially selected 34 trials, 9 studies (26%) were excluded because the ESMO-MCBS could not be applied. It would be beneficial if the authors could provide a more detailed explanation in the text regarding why the ESMO-MCBS could not be applied in these 9 studies.

Response: Thank you very much for your important comments. The 9 studies included negative results (e.g. KAITLIN and NCT02568839), while the ESMO-MCBS can only be applied to studies showing statistically significant differences. Therefore, we excluded these 9 studies. Also, we have removed the Figure 1 and revised the Methods section.

11) ESMO-MCBS scorecards are only published for drugs that are approved by the FDA and EMA. Can the authors clarify if the 8 studies that do not have scorecards are because they are only approved by jurisdictions other than the European and American ones? In the case of one study (trastuzumab deruxtecan in colorectal cancer), it could be because it is not approved.

Response: Thank you for your constructive comments. It is really true as Reviewer suggested that ESMO-MCBS scorecards are only published for drugs that are approved by the FDA and EMA. A total of 7 publications do not have scorecards. The drugs were not approved based on 4 trials (TH3RESA, DESTINY-Breast02, DESTINY-Lung01 and DESTINY-CRC01), and this was inconsistent with the methods in our manuscript. We have revised the methods section. Another 3 trials (NCT02422979, NCT03556345 and NCT03507166) do not have scorecards because they are approved by jurisdictions other than the EMA and US FDA.

12) Limitations: While the manuscript mentions limitations, it would benefit from a more comprehensive discussion of potential biases and how they might affect the study's conclusions. For example, among the 13 therapeutic ADCs approved by the FDA, 7 belong to the hematological field. This is a major limitation that should be at least mentioned.

Response: We appreciate your valuable comments. We have added it and revised the Limitations section (Page 11, Lines 263-267). We also pasted here for your review:

We limited our analysis to solid tumors and excluded drugs approved to treat hematologic malignancies. When this study was completed in June 2023, the haematology-specific version has yet to be published. We will further assess the clinical benefits of ADCs for hematologic malignancies according to the currently published ESMO-MCBS:H v 1.0.

13) Interpretation of Results: The discussion could provide a deeper analysis of the findings and their clinical implications in relationship to ADCs.

Response: We appreciate your valuable comments. This study has several implications. Firstly, this study showed clinical benefits for limited ADCs in solid tumors, suggesting that subsequent clinical trials on the treatment of solid tumors with ADCs should follow the cases with meaningful clinical benefit. Secondly, value frameworks can help not only identify drugs with low or uncertain clinical benefit that should be targeted for price negotiations, but also therapies with evidence of higher clinical benefit to improve access to benefit drugs, thereby contributing to patient-centred cancer treatment goals. Finally, the ADCs with HRQoL improvement showed clinical benefit, suggesting that HRQoL should be paid sufficient attention in clinical trials and clinical treatment strategies (Page 11, Lines 254-262).

14) Conclusion: The conclusion section is somewhat abrupt and could be expanded to provide a more comprehensive summary of the study's significance, public health implications and potential future directions

Response: We accept your brilliant suggestion. We have revised the Conclusion section (Page 12, Lines 278-285). We also pasted it here for your review:

In conclusion, this is the first study that used ESMO-MCBS tool to assess clinical benefit of ADCs across several solid cancers. ESMO-MCBS are important tools for assessing clinical benefit of cancer drugs, although not all drugs met the meaningful threshold. Those therapeutic regimens with

improved HRQoL showed clinical value, suggesting that clinical trials and clinical treatment strategies should pay more attention to HRQoL. Besides, a more detailed approach to toxicity penalties that accounts for adverse event is required, as well as an adjusted scoring for those studies with HRQoL.

VERSION 2 – REVIEW

REVIEWER	Grossmann, Nicole Austrian Inst Hlth Technol Assessment
REVIEW RETURNED	25-Oct-2023

GENERAL COMMENTS	Dear Authors, I hope this message finds you well. I want to express my gratitude for your diligent efforts in revising the paper in accordance with the reviewers' suggestions. I have thoroughly examined the updated manuscript and commend you for your commitment to addressing the comments and concerns. Overall, the changes made have significantly enhanced the paper's quality. Upon re-reading your paper, I've recognized that, for the sake of completeness, it would be beneficial to include the ESMO-MCBS calculations for ADCs (Antibody-Drug Conjugates) related to hematological cancers. Incorporating this additional ESMO-MCBS scores would provide valuable insights into the comprehensive landscape of ADCs.
--

REVIEWER	Tibau, Ariadna Sant Pau and Universitat Autònoma de Barcelona
REVIEW RETURNED	28-Oct-2023

GENERAL COMMENTS	First of all, congratulate the authors for the corrections to the article. I add some few suggestions:  1. The study is at some points still difficult to follow regarding numbers. For example, for trastuzumab emtansine, the authors mention in the results section that they reviewed 4 studies (L125), but only two studies appears in table 1, and to my knowledge, it is a drug that only has 2 indications. 2. In the results section, the authors do not merely indicate the results in a descriptive manner but make personal statements such as "we think" (for example page 33 L165); my suggestion is that they express personal opinions only in the discussion section. 3. An important limitation of the study is that the authors have conducted a PubMed search for articles instead of searching the websites of different regulatory agencies for approved indications (and supporting articles). This detail weakens the solidity of the article. Additionally, statements such as "In our study, several ADCs have been approved by local agencies in China and Japan" (L165) without providing the number create some confusion. In Table 1, only 3 indications are listed (2 approved in China and 1 in
--

	Japan). I recommend specifying which ADCs have been approved in each jurisdiction; a supplementary table would be helpful as well as to recognize this limitation in the discussion section. 4. An interesting part of the study is the observation of discrepancies in 5 out of 12 (42%) trials (Table 2). It would be interesting for the authors to better report the reasons behind these discrepancies, as they only explain this in two studies (TROPiCS-02 and EMILIA). 5. Finally, in Figure 1 (flowchart), the final number of evaluated studies should match the number analyzed in the study.
--	---

VERSION 2 – AUTHOR RESPONSE

Reviewer: 1

Dr. Nicole Grossmann, Austrian Inst Hlth Technol Assessment

Comments to the Author:

Dear Authors,

I hope this message finds you well. I want to express my gratitude for your diligent efforts in revising the paper in accordance with the reviewers' suggestions. I have thoroughly examined the updated manuscript and commend you for your commitment to addressing the comments and concerns. Overall, the changes made have significantly enhanced the paper's quality.

Upon re-reading your paper, I've recognized that, for the sake of completeness, it would be beneficial to include the ESMO-MCBS calculations for ADCs (Antibody-Drug Conjugates) related to hematological cancers. Incorporating this additional ESMO-MCBS scores would provide valuable insights into the comprehensive landscape of ADCs.

Response: Thank you for your constructive comments. Our field is not hematologic malignancies and we have little expertise in hematological cancers. Therefore, we invited Prof. Jesus M Hernandez (Servicio de Hematología, Instituto de Investigación Biomédica de Salamanca; e-mail: jmhr@usal.es) to guide us to determine which Form to use for scoring the 7 ADCs approved for hematological cancers. Unfortunately, we cannot contact him now. We also evaluated the ESMO-MCBS calculations for ADCs related to hematological cancers ourselves as follows; however, we suspected the correctness and do not recommend adding this section to the paper. If Prof. Hernandez contacts us in the future, we will consider further publication on the ESMO-MCBS calculations for ADCs related to hematological cancers

Trial name	Intervention group	PO	PO ctrl group	PO gain	PO HR (95% CI)	Adjustments	ESMO-MCBS:H
------------	--------------------	----	---------------	---------	----------------	-------------	-------------

ALFA-0701	GO plus daunorubicin/ cytarabine	EFS (OS improved)	19.2 mo	14.8 mo	0.69 (0.49–0.98)	NA	4 (form 2a)
AML-19	GO	OS	3.6 mo	1.3 mo	0.69 (0.53–0.90)	NA	1 (form 2a)
MyloFrance-1	GO	ORR	-	-	-	NA	3 (form 3)
ECHELON-1	BV plus AVD	PFS (OS improved)	6-year OS: 89.4%	6-year OS: 4.5%	0.59 (0.40–0.88)	NA	2 (form 2a)
AETHERA	BV	PFS	24.1 mo	18.8 mo	0.57 (0.40–0.81)	NA	3 (form 2b)
SG035-0003	BV	ORR	-	-	-	NA	3 (form 3)
ECHELON-2	BV plus CHP	PFS	20.8 mo	27..4 mo	0.71 (0.54–0.93)	≥10% improvement in 3-year PFS	2 (form 2b)
SG035-0004	BV	ORR	-	-	-	NA	3 (form 3)
ALCANZA	BV	ORR4	12.5%	43.8%	-	NA	2 (form 2c)
INO-VATE ALL	IO	CR and OS	6.7 mo	1.0 mo	0.77 (0.58–1.03)	NA	1 (form 2a)
Study 1053	MP	CR	-	-	-	NA	3 (form 3)
POLARIX	PV	PFS	2-year PFS: 70.2%	2-year PFS: 6.5%	0.73 (0.57–0.95)	NA	NA
GO29365	PV plus BG	CR (OS improved)	4.7 mo	7.7 mo	0.42 (0.24–0.75)	NA	4 (form 2a)
DREAMM-2	BM	ORR	-	-	-	NA	2 (form 3)
LOTIS-2	LT	ORR	-	-	-	NA	2 (form 3)

Reviewer: 4

Dr. Ariadna Tibau, Sant Pau and Universitat Autònoma de Barcelona

Comments to the Author:

First of all, congratulate the authors for the corrections to the article.

I add some few suggestions:

1. The study is at some points still difficult to follow regarding numbers. For example, for trastuzumab emtansine, the authors mention in the results section that they reviewed 4 studies (L125), but only two studies appears in table 1, and to my knowledge, it is a drug that only has 2 indications.

Response: We apologize for this confusion of the concept of indications/approvals, trials and publications. It is true as your comment that trastuzumab emtansine only has 2 indications. However, we included 4 publications on the trastuzumab emtansine. Apart from the 2 publications led to approval, we also included another 2 publications on the final OS analysis and patient-reported outcomes. To avoid this confusion, we have revised the related sentences. We also pasted it here for your review:

A total of 17 publications (9 randomized controlled trials and 8 single-arm trials) leading to approval for solid cancers were finally identified, including 2 for trastuzumab emtansine, 1 for enfortumab vedotin, 6 for trastuzumab deruxtecan, 3 for sacituzumab govitecan, 1 for cetuximab saratolacan, 2 for disitamab vedotin, 1 for tisotumab vedotin and 1 for mirvetuximab soravtansine. The approved ADCs covered 7 solid cancer types, including breast cancer, urothelial carcinoma, gastric cancer, non-small cell lung cancer, head-and-neck squamous cell carcinoma, cervical cancer and ovarian cancer (table 1). The publications on updated efficacy or QoL data were also included for ESMO-MCBS score adjustment.

2. In the results section, the authors do not merely indicate the results in a descriptive manner but make personal statements such as "we think" (for example page 33 L165); my suggestion is that they express personal opinions only in the discussion section.

Response: Thank you very much for your important comments. We have removed it and double checked our whole context.

3. An important limitation of the study is that the authors have conducted a PubMed search for articles instead of searching the websites of different regulatory agencies for approved indications (and supporting articles). This detail weakens the solidity of the article. Additionally, statements such as "In our study, several ADCs have been approved by local agencies in China and Japan" (L165) without providing the number create some confusion. In Table 1, only 3 indications are listed (2 approved in China and 1 in Japan). I recommend specifying which ADCs have been approved in each jurisdiction; a supplementary table would be helpful as well as to recognize this limitation in the discussion section.

Response: We appreciate your important comments. We have searched the websites of different regulatory agencies for approved indications. We have also added it in manuscript. In addition, we have also added the approved year and agencies in Table 1 to specify which ADCs have been approved in each jurisdiction.

4. An interesting part of the study is the observation of discrepancies in 5 out of 12 (42%) trials (Table 2). It would be interesting for the authors to better report the reasons behind these discrepancies, as they only explain this in two studies (TROPiCS-02 and EMILIA).

Response: We are grateful for the suggestion. As we mentioned in the manuscript that different scores mostly due to integration of more recent data, we have added more detailed reasons for the discrepancies in these 5 trials.

5. Finally, in Figure 1 (flowchart), the final number of evaluated studies should match the number analyzed in the study.

Response: We accept your brilliant suggestion. We have removed the Figure 1 in the previous round of revision, because the selection procedure in Figure 1 was inconsistent and we directly included the trials leading to approval.

VERSION 3 – REVIEW

REVIEWER	Tibau, Ariadna Sant Pau and Universitat Autònoma de Barcelona
REVIEW RETURNED	18-Feb-2024
GENERAL COMMENTS	The paper has been revised to address many of the issues raised by me as a reviewer. I have no further comments. Congratulations to the authors!

VERSION 3 – AUTHOR RESPONSE